# Pseudo Label-Based Semi-Supervised Learning for Abdominal Organ and Cancer Segmentation in CT Image With Partial Labeled Data

Panlong Xu[1][0000-0001-7200-1238], Zhijian Li[1][0009-0008-6050-8216], and Weiping Liu[1]

Imaging and Navigation Research Center, Shanghai Microport
Medbot(Group)Co.,Ltd., Shanghai, China
`{xupl}@microport.com`

**Abstract.** Abdominal multi-organ and pan-tumor segmentation in CT image plays a critically important role in preoperative planning, intraoperative navigation, and postoperative assessment for surgical procedures.In this study, we propose a semi-supervised learning approach using nnU-Net on the FLARE2023 competition dataset. Our methodology involves training an initial model on fully annotated data, followed by inference on partially annotated data to generate pseudo-labels, and subsequently training a final model using these pseudo-labeled data. To optimize computational efficiency, we adopt a parameter-efficient model with a reduced number of parameters. By leveraging the availability of both labeled and unlabeled data, our approach aims to enhance the performance of the nnU-Net model while maintaining a reasonable computational cost. Ultimately, our trained small nnU-Net achieved significant results on a validation set of 100 samples, with a dice coefficient of 0.8854 for multi-organ segmentation and 0.4186 for tumor segmentation. Moreover, the average inference time of the model was only 18 seconds.

**Keywords:** Semi-supervised learning · Image segmentation · Pseudo label.

## 1 Introduction

The FLARE2023 challenge aims to promote the development of universal organ and tumor segmentation in abdominal CT scans, which is an extension of FLARE2021 and FLARE2022 challenge. The participants should develop segmentation algorithm which enable segment 13 organs (liver, spleen, pancreas, right kidney, left kidney, stomach, gallbladder, esophagus, aorta, inferior vena cava, right adrenal gland, left adrenal gland, and duodenum) and one tumor class with all kinds of cancer types (such as liver cancer, kidney cancer, stomach cancer, pancreas cancer, colon cancer) in abdominal CT scans.

Abdominal organ and tumor segmentation hold significant clinical importance in several aspects. Firstly, accurate segmentation of abdominal organs

allows for precise identification and analysis of specific structures, aiding in surgical planning by providing detailed information about the spatial relationships between organs. This assists surgeons in determining the optimal surgical approach and reducing the risk of complications during the procedure.Furthermore, tumor segmentation plays a crucial role in the diagnosis, treatment planning, and evaluation of cancer patients. By accurately delineating tumor boundaries, clinicians can assess tumor size, location, and response to therapy. This information guides treatment decisions, such as determining the extent of surgical resection and predicting prognosis.

However, multi-organ segmentation and pan-tumor segmentation face several challenges. Firstly, variations in organ shape, size, and appearance across different individuals and disease states make accurate segmentation challenging. Secondly, the presence of overlapping structures and ambiguous boundaries between organs or tumors adds difficulty to the segmentation task. Finally, image artifacts, noise, and limited image resolution can affect the quality of segmentation results.

The nnU-Net [6] segmentation framework has proven effective in addressing the challenges mentioned due to its ability to analyze the fingerprint features of training data. By understanding the unique characteristics of the data, nnU-Net can adapt the network structure complexity and preprocessing strategy accordingly. This adaptability enables the framework to handle variations in organ shapes, sizes, and appearances, as well as cope with ambiguous boundaries and image artifacts. As a result, nnU-Net can provide accurate and robust segmentation results for multi-organ and pan-tumor segmentation tasks.

In this study, we propose a semi-supervised learning approach based on nnU-Net to solve the abdominal multi-organ and pan-tumor segmentation problem in CT images. Our methodology involves training an initial model on fully annotated data, followed by inference on partially annotated data to generate pseudo-labels, and subsequently training a final model using these pseudo-labeled data. To optimize computational efficiency, we adopt a parameter-efficient model with a reduced number of parameters. By leveraging the availability of both labeled and unlabeled data, our approach aims to enhance the performance of the nnU-Net model while maintaining a reasonable computational cost.

## 2   Method

The FLARE2023 challenge provide the largest abdomen CT dataset. The training set includes 4000 3D CT scans from 30+ medical centers. 2200 cases have partial labels and 1800 cases are unlabeled.Despite the availability of a large training dataset, a statistical analysis revealed severe class imbalance in the annotated dataset. Notably, among the 2,200 annotated examples, none included annotations for all 14 classes. The graph below illustrates the distribution of annotations for each class in the incomplete dataset, indicating significantly fewer annotations for classes 5-12 compared to others. To address this issue, we em-

ployed a semi-supervised learning approach based on pseudo-labeling to iteratively train the segmentation model. The training process involved five stages:

1. In the first stage, we trained a segmentation model using 222 examples annotated for classes 1-13.

2. In the second stage, we selected 597 examples annotated for classes 1-4, 13, and 14. Using the segmentation model from the first stage, we inferred the unannotated classes and trained a 14-class segmentation model.

3. The third stage involved inferring tumor pseudo-labels using the model from the second stage on the training data from the first stage. The combined dataset of 819 annotated examples was then used to train the segmentation model.

4. In the fourth stage, we utilized the model from the third stage to inference the unannotated class labels for the remaining 1,200 examples, and mixed the entire dataset of 2,200 examples for training.

5. Finally, in the fifth stage, we inferred the labels for the remaining 1,800 unannotated examples using the model obtained from the fourth stage. The model was then further trained through the hybrid training process to obtain the final segmentation model.

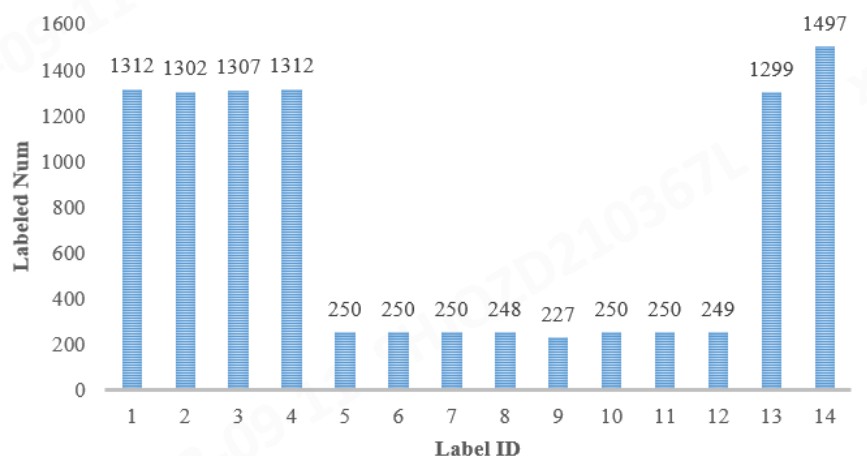

**Fig. 1.** Statistics on the annotated data for each class in the training set. The horizontal axis represents the IDs of the 14 different classes, while the vertical axis represents the number of annotated samples.

## 2.1 Preprocessing

All data preprocessing follows the original nnU-Net framework. Firstly, the raw images are cropped to remove contiguous regions with pixel values of 0, although such cases do not exist in real CT images. Secondly, the images are resampled

according to the predetermined spacing, as shown in Table 1 and Table 2. The input spacing for larger model is smaller than that for smaller model. Finally, the data is normalized, with two threshold values of 0.05 and 0.95 obtained from pixel value statistics used for truncation.

## 2.2  Proposed Method

As shown in Figure 2, our proposed method contains five training stages. Meanwhile, two different size 3D nnU-Net were applied to train different models.

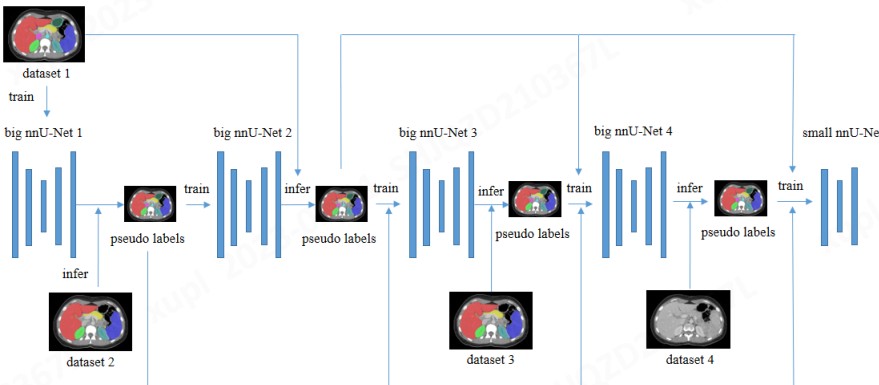

**Fig. 2.** Pipeline of our proposed training strategy.

The big nnU-Net model is characterized by a wider and deeper network structure and a higher input resolution. On the contrary, the small nnU-Net model has a narrower and shallower network structure, along with a lower input resolution. The differences in their network structures can be observed from Table 1 and Table 2. This training strategy is inspired by the approach described in [5]. During the initial training phase with partially labeled data, we aim to obtain more reliable pseudo-labels. As for the final network training, we balance the inference speed, memory consumption, and segmentation accuracy by reducing the complexity of the segmentation model.

**Table 1.** Big nnU-Net network structure.

| Settings | Value |
|---|---|
| channels in the first stage | 32 |
| convolution number per stage | 3 |
| downsampling times | 5 |
| input spacing | (2.5, 0.8, 0.8) |
| test time augmentation | yes |

**Table 2.** Small nnU-Net network structure.

| Settings | Value |
|---|---|
| channels in the first stage | 16 |
| convolution number per stage | 2 |
| downsampling times | 4 |
| input spacing | (4.0, 1.2, 1.2) |
| test time augmentation | no |

Loss function: we use the summation between Dice loss and cross-entropy loss because compound loss functions have been proven to be robust in various medical image segmentation tasks [7].

Regarding the training process, as mentioned before, the entire dataset suffers from severe class imbalance. Through statistical analysis of the annotated data, we have proposed a hierarchical training strategy, progressing from easy to difficult and gradually increasing the number of classes. This strategy is illustrated in Figure 2. Firstly, we select dataset1, which consists of 222 examples annotated with class labels ranging from 1 to 13. With this set of data, we train a large nnU-Net, which named big nnU-Net 1. In the second step, we further select dataset2, comprising 597 examples annotated with classes 1 to 4, 13, and 14 (a total of 6 classes). To supplement the missing 8 classes, we utilize the inference results from big nnU-Net 1 and then train another large nnU-Net model, called big nnU-Net 2. The third step involves using big nnU-Net 2 to infer the missing tumor annotation in dataset1, then combined with dataset2 and train big nnU-Net 3. For the fourth step, the remaining 1381 partially annotated data samples form dataset3. We use big nnU-Net 3 to infer the missing labels, obtain pseudo-labels, and mix them with the rest of the data to train big nnU-Net 4.

To fully utilize the remaining unlabeled data and strike a balance between inference speed and memory consumption, in the final stage of training, we employ 2200 unlabeled examples, namely dataset4, to train the small nnU-Net. We don't used the pseudo labels generated by the FLARE21 winning algorithm [5] and the best-accuracy-algorithm [14].

In order to improve inference speed and reduce resource consumption, on one hand, we have reduced the model complexity and the size of input patches. On the other hand, we have adopted the same sliding window strategy as described in [5].

### 2.3 Post-processing

During the post-processing stage, we experimented with connected component operations but found that they hardly improved the final results. As a result, we ultimately decided not to employ any post-processing operations.

## 3    Experiments

### 3.1    Dataset and evaluation measures

The FLARE 2023 challenge is an extension of the FLARE 2021-2022 [9][10], aiming to aim to promote the development of foundation models in abdominal disease analysis. The segmentation targets cover 13 organs and various abdominal lesions. The training dataset is curated from more than 30 medical centers under the license permission, including TCIA [2], LiTS [1], MSD [13], KiTS [3,4], and AbdomenCT-1K [11]. The training set includes 4000 abdomen CT scans where 2200 CT scans with partial labels and 1800 CT scans without labels. The validation and testing sets include 100 and 400 CT scans, respectively, which cover various abdominal cancer types, such as liver cancer, kidney cancer, pancreas cancer, colon cancer, gastric cancer, and so on. The organ annotation process used ITK-SNAP [15], nnU-Net [6], and MedSAM [8].

The evaluation metrics encompass two accuracy measures—Dice Similarity Coefficient (DSC) and Normalized Surface Dice (NSD)—alongside two efficiency measures—running time and area under the GPU memory-time curve. These metrics collectively contribute to the ranking computation. Furthermore, the running time and GPU memory consumption are considered within tolerances of 15 seconds and 4 GB, respectively.

### 3.2    Implementation details

**Environment settings** The development environments and requirements are presented in Table 3. The training protocols of big nnU-Net and small nnU-Net are listed in Table 4 and  5 respectively. We adopt data augmentation of additive brightness, gamma, rotation, scaling, and elastic deformation on the fly during training.

**Table 3.** Development environments and requirements.

| | |
|---|---|
| System | Ubuntu 20.04.3 LTS |
| CPU | AMD EPYC 7643 48-Core Processor@1.50GHz |
| RAM | 504GB |
| GPU (number and type) | One NVIDIA A100 40G |
| CUDA version | 11.6 |
| Programming language | Python 3.8 |
| Deep learning framework | torch 1.12 |

## 4    Results and discussion

### 4.1    Quantitative results on validation set

After multiple rounds of iterative training and hierarchical learning, the big nnU-Net4 model has achieved good performance in segmentation. The average

**Table 4.** Big nnU-Net training protocols.

| | |
|---|---|
| Network initialization | "He" normal initialization |
| Batch size | 2 |
| Patch size | 48×224×224 |
| Total epochs | 1000 |
| Optimizer | SGD with nesterov momentum ($\mu = 0.99$) |
| Initial learning rate (lr) | 0.01 |
| Lr decay schedule | Poly learning rate policy |
| Training time | 24 hours |
| Loss function | Dice loss and cross entropy loss |
| Number of model parameters | 82M[1] |
| Number of flops | 776G[2] |
| $CO_2$eq | 34 Kg[3] |

**Table 5.** Training protocols for the small nnU-Net.

| | |
|---|---|
| Network initialization | "He" normal initialization |
| Batch size | 2 |
| Patch size | 32×128×192 |
| Total epochs | 1000 |
| Optimizer | SGD with nesterov momentum ($\mu = 0.99$) |
| Initial learning rate (lr) | 0.01 |
| Lr decay schedule | Poly learning rate policy |
| Training time | 12 hours |
| Number of model parameters | 5.4M[4] |
| Number of flops | 136G[5] |
| $CO_2$eq | 11 Kg[6] |

**Table 6.** Quantitative evaluation results on the public 50 validation cases and 100 online validation cases.

| Target | Public Validation | | Online Validation | | Testing | |
|---|---|---|---|---|---|---|
| | DSC(%) | NSD(%) | DSC(%) | NSD(%) | DSC(%) | NSD (%) |
| Liver | 97.27±0.48 | 98.88±1.44 | 97.23 | 98.91 | | |
| Right Kidney | 94.58±7.79 | 96.10±7.34 | 93.83 | 95.47 | | |
| Spleen | 96.38±1.07 | 98.35±2.35 | 96.61 | 98.80 | | |
| Pancreas | 85.50±5.44 | 97.00±3.56 | 84.37 | 96.22 | | |
| Aorta | 95.69±1.25 | 98.88±1.83 | 95.80 | 98.85 | | |
| Inferior vena cava | 93.97±1.85 | 96.70±2.64 | 93.77 | 96.16 | | |
| Right adrenal gland | 81.89±5.08 | 95.04±2.73 | 80.84 | 94.46 | | |
| Left adrenal gland | 78.82±6.00 | 93.18±4.88 | 78.35 | 92.43 | | |
| Gallbladder | 79.34±24.71 | 78.00±25.19 | 79.79 | 78.12 | | |
| Esophagus | 81.43±14.71 | 92.91±14.67 | 82.06 | 93.90 | | |
| Stomach | 91.93±3.14 | 96.97±4.42 | 92.46 | 97.48 | | |
| Duodenum | 81.55±7.19 | 94.83±5.32 | 82.47 | 95.38 | | |
| Left kidney | 93.37±9.80 | 94.18±12.15 | 93.41 | 94.93 | | |
| Tumor | 48.40±34.35 | 39.18±30.50 | 41.86 | 33.81 | | |
| Average | 85.72±17.82 | 90.73±19.46 | 85.17 | 90.35 | | |

Dice coefficient for organ segmentation is 0.895, and for tumor segmentation, it is 0.447. The tumor segmentation metric ranked ninth on the validation leaderboard. Figure 3 presents a comparative analysis of the average Dice coefficient achieved by three models during the training process on the online validation dataset.

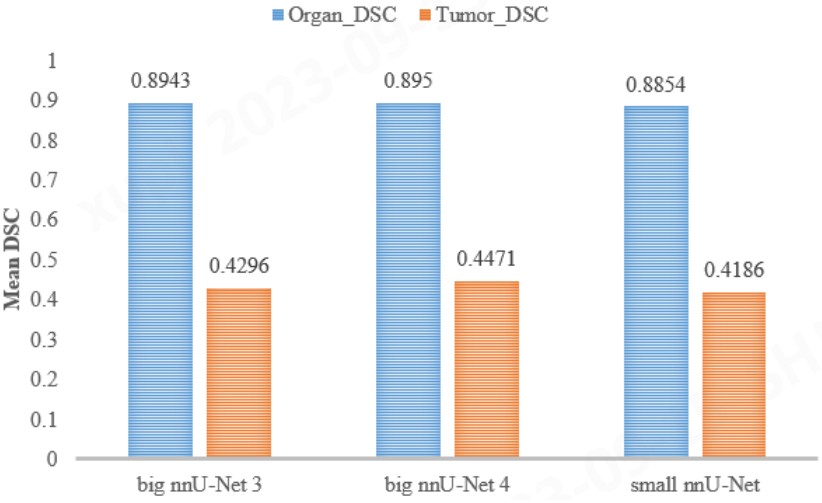

**Fig. 3.** Comparison of different models on validation mean Dice metric.

As for the final small nnU-Net model, the average Dice coefficient for organ segmentation on the validation set is 0.885, and for tumor segmentation, it is 0.419. Although there is a slight decrease in segmentation performance, the inference time cost and GPU memory consumption have been significantly reduced. The quantitative evaluation results of the small nnU-Net on the validation set are shown in Table 6.

### 4.2   Qualitative results on validation set

Despite the reduced complexity of the small nnU-Net model, the inclusion of a large amount of unlabeled data in the training process allows the model to maintain good segmentation performance on various organs. Figure4 illustrates two well-segmented cases, demonstrating the model's ability to capture organ edges and details accurately. However, due to the decrease in resolution, the segmentation performance of the model is more noticeably affected on smaller anatomical structures, particularly tumors. Figure5 displays two cases where the segmentation results are less satisfactory.

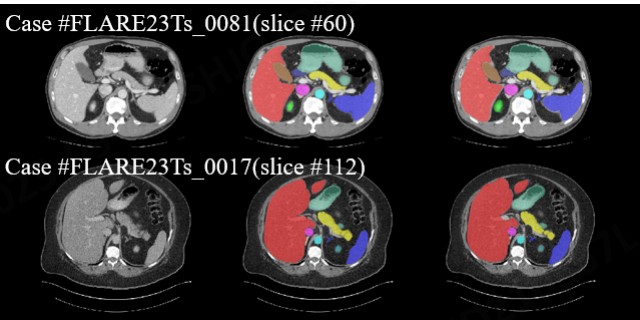

**Fig. 4.** Qualitative results of the small nnU-Net on two easy cases.

### 4.3   Segmentation efficiency results on validation set

We build our small nnU-Net with an efficient inference strategy as a docker image for final submission. In Table7, we report the efficiency evaluation results on the FLARE2023 organizer's computer server with GPU NVIDIA QUADRO RTX5000.

### 4.4   Results on final testing set

### 4.5   Limitation and future work

Although our final model has shown promising performance in terms of inference speed and GPU memory consumption, there is still considerable room for

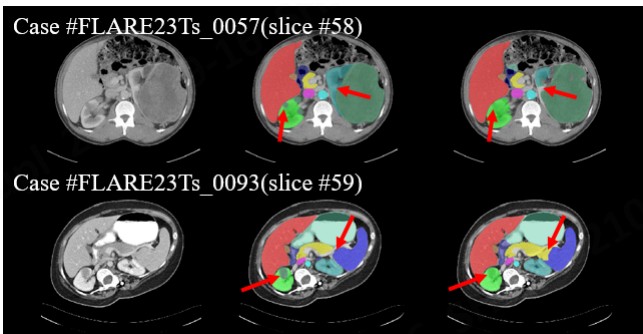

**Fig. 5.** Qualitative results of the small nnU-Net on two hard cases.

**Table 7.** Efficiency evaluation results of our submitted docker. All metrics reported are the average values on 20 validation cases.

| Time | GPU Memory | AUC GPU Time | CPU Utilization | AUC CPU Time |
|------|-----------|--------------|-----------------|--------------|
| 18.5s | 2532MiB | 18466 | 63.2% | 357 |

improvement in its segmentation performance. In the future, we will explore more semi-supervised learning techniques, particularly deep learning methods based on auto-encoders. By extracting high-level semantic features from a large amount of data and transferring the learned feature descriptors to downstream segmentation tasks, we aim to enable the segmentation model to converge faster and achieve higher accuracy.

## 5   Conclusion

In this paper, we propose a semi-supervised training strategy based on nnU-Net. Specifically, we adopt a hierarchical learning approach to leverage both partially labeled and unlabeled data. We progressively train the model from easy to difficult samples. Additionally, to accelerate the model's inference speed, we reduce its complexity. We believe that our approach can provide valuable insights and inspiration for other researchers in this field.

**Acknowledgements**  The authors of this paper declare that the segmentation method they implemented for participation in the FLARE 2023 challenge has not used any pre-trained models nor additional datasets other than those provided by the organizers. The proposed solution is fully automatic without any manual intervention. We thank all the data owners for making the CT scans publicly available and CodaLab [12] for hosting the challenge platform.

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

**Table 8.** Checklist Table. Please fill out this checklist table in the answer column.

| Requirements | Answer |
| --- | --- |
| A meaningful title | Yes |
| The number of authors ($\leq 6$) | 3 |
| Author affiliations, Email, and ORCID | Yes |
| Corresponding author is marked | Yes |
| Validation scores are presented in the abstract | Yes |
| Introduction includes at least three parts: background, related work, and motivation | Yes |
| A pipeline/network figure is provided | 2 |
| Pre-processing | 3 |
| Strategies to use the partial label | 5 |
| Strategies to use the unlabeled images. | 5 |
| Strategies to improve model inference | 5 |
| Post-processing | 5 |
| Dataset and evaluation metric section is presented | 6 |
| Environment setting table is provided | 2 |
| Training protocol table is provided | 4,5 |
| Ablation study | 8 |
| Visualized segmentaiton example is provided | 9,10 |
| Limitation and future work are presented | Yes |
| Reference format is consistent. | Yes |