# OpenReview forum: "Pseudo Label-Based Semi-Supervised Learning for Abdominal Organ and Cancer Segmentation in CT Image With Partial Labeled Data"
_MICCAI.org/2023/FLARE — Submitted to FLARE 2023_

### Official Review · Reviewer_Xh1c · 2023-09-22
**Sufficient but the details still need to be described more clearly**

**Rating:** 6
**Confidence:** 4

**Review:**

This paper uses the nn-UNet framework to iteratively train partial label data and supplement annotations in order to create a fully annotated dataset. The efficiency of the inference phase is optimized by training a lightweight nn-UNet with a fully annotated dataset.

Comment:
1. The dataset division needs to be clearer during iteration. If the reader has not understood the dataset, it will be in chaos. Include details of the dataset used in each phase should in Figure 2 and the various datasets mentioned in the following descriptions maintain their designated names consistently.

---

### Official Review · Reviewer_DHY8 · 2023-09-26
**It's complete, but it needs a little more.**

**Rating:** 7
**Confidence:** 4

**Review:**

Summary
This paper presents an extension of nn-UNet for semi-supervised learning. The author utilizes the nn-UNet framework to train a large model using labeled data and inferences the unlabeled data. Subsequently, a lightweight nn-UNet is trained using a pseudo-labeled dataset consisting of 14 classes of labels. This approach aims to optimize the efficiency of the inference stage while achieving satisfactory results.

Comments
1.This paper does not include a comparison of the running time and efficiency between the big nn-UNet and small nn-UNet models. Additionally, it does not provide a comparison of the running efficiency when using raw data of different sizes.
2.In the method section, it would be beneficial to provide a clearer explanation of the data division during the entire training process. Clarifying how the data is divided and utilized would enhance the reproducibility and understanding of the proposed method.
3.If the author can provide open source code, it will be very beneficial.

---

### Official Review · Reviewer_dTTW · 2023-09-30
**Great paper, but a little more information needed.**

**Rating:** 7
**Confidence:** 4

**Review:**

Pros:
1. The proposed method accurately and efficiently segments abdominal organs and tumors. In online validation, the DSC values for organ and tumor segmentation are 88.54% and 41.86%, respectively. The inference time was an average of 18 seconds, and the area under the GPU memory time curve was an average of 18466 MB.

Cons:
1. Abstract: Please add information about the area under GPU memory-time curve.
2. It would be highly beneficial if the authors could provide open-source code.
3. Step 4 on Page 3: " remaining 1,200 examples" should be " remaining 1,381 examples."
4. Fig.2: Please add more information in the figure annotation to explain the process of this pipeline.
5. Page 5: Replacing "big nnU-Net 1" with "M1" or other mathematical symbols helps to improve recognition, and the same applies to Figure 2 or other places where it is used.
6. Page 5: Replacing "dataset1" with "D1" or other mathematical symbols helps to improve recognition, and the same applies to Figure 2 or other places where it is used.
7. Page 5: "2200 unlabeled examples" should be "1800 unlabeled examples". "FLARE21" should be "FLARE22".
8. Please harmonize the expression of numbers, either one or the other, like 1200 or 1,200.
9. Section 3.1: "aiming to aim to" should be "aiming to".
10. Please add a table about " Quantitative evaluation of segmentation efficiency in terms of the running them and GPU memory consumption" from the official template.

---

### Official Review · Reviewer_FZ7t · 2023-10-04
**It's good, but there are a few things missing**

**Rating:** 6
**Confidence:** 4

**Review:**

- Pros
    - The proposed hierarchical learning approach to leverage both partially labeled and unlabeled data is detailed and shows significant performance.
- Cons
    - Abstract: No results on GPU consumption.
    - Introduction: No references other than nnU-Net.
    - Results: No ablation study for unlabeled data.
    - Results: Missing quantitative evaluation of segmentation efficiency for the specified 8 cases from the template.

---

### Public Comment · ~PENGJU_LYU1 · 2023-11-26
**add test results**

please fill in the final test results.

---

### Decision · Program_Chairs · 2023-10-24

**Decision:**

Reject

**Comment:**

The authors didn't make responses to the valuable review comments.